# HalluEntity: Benchmarking and Understanding Entity-Level Hallucination Detection

**Min-Hsuan Yeh**                                    *samuelyeh@cs.wisc.edu*
*University of Wisconsin-Madison*

**Max Kamachee**                                     *kamachee@cs.wisc.edu*
*University of Wisconsin-Madison*

**Seongheon Park**                          *seongheon_park@cs.wisc.edu*
*University of Wisconsin-Madison*

**Yixuan Li**                                         *sharonli@cs.wisc.edu*
*University of Wisconsin-Madison*

**Reviewed on OpenReview:** https://openreview.net/forum?id=494k7e9R5D

## Abstract

To mitigate the impact of hallucination nature of LLMs, many studies propose detecting hallucinated generation through uncertainty estimation. However, these approaches predominantly operate at the sentence or paragraph level, failing to pinpoint specific spans or entities responsible for hallucinated content. This lack of granularity is especially problematic for long-form outputs that mix accurate and fabricated information. To address this limitation, we explore *entity-level hallucination detection*. We propose a new data set, HALLUENTITY, which annotates hallucination at the entity level. Based on the dataset, we comprehensively evaluate uncertainty-based hallucination detection approaches across 17 modern LLMs. Our experimental results show that uncertainty estimation approaches focusing on individual token probabilities tend to over-predict hallucinations, while context-aware methods show better but still suboptimal performance. Through an in-depth qualitative study, we identify relationships between hallucination tendencies and linguistic properties and highlight important directions for future research.

**HalluEntity:** https://huggingface.co/datasets/samuelyeh/HalluEntity

## 1 Introduction

How can we trust the facts generated by large language models (LLMs) when even a single hallucinated entity can distort an entire narrative? While LLMs have revolutionized text generation in various domains, from summarization to scientific writing (Liang et al., 2024), their tendency to produce hallucinations—factually incorrect or unsupported content—remains a critical challenge (Xu et al., 2024). This issue is particularly concerning in high-stakes applications, such as medical diagnostics (Chen et al., 2024b), legal document drafting (Lin & Cheng, 2024), and news generation (Odabaşı & Biricik, 2025), where inaccurate information can cause harm to individuals and erosion of public trust. Detecting hallucinations is therefore a critical step toward ensuring the responsible deployment of LLMs.

Various approaches have been proposed to detect hallucinations (Luo et al., 2024), with uncertainty-based methods emerging as a promising direction (Zhang et al., 2023a). However, current uncertainty-based hallucination detection approaches mainly operate at the sentence or paragraph level, classifying the entire generation as either hallucinated or correct. While this provides a coarse-grained assessment of factuality, it lacks the granularity needed to pinpoint which specific spans or entities contribute to hallucination. This

limitation is particularly problematic for long-form text, where both accurate and hallucinated information frequently coexist. For example, a generated response about a historical event might correctly state the date but fabricate details about the individuals involved, necessitating finer-grained detection.

To address these limitations, we present a first systematic exploration of ***entity-level hallucination detection*** by introducing a benchmark dataset, extensively evaluating uncertainty-based detection methods on this benchmark, and analyzing their strengths and limitations in identifying hallucinated entities. Specifically, we begin by constructing a benchmark for entity-level hallucination detection, HALLUENTITY, which fills in the critical blank for the field. Constructing such a dataset is challenging due to the labor-intensive nature of entity-level annotation, which requires annotators to segment meaningful entities and verify their factuality against reliable sources one by one. To overcome this, we develop a systematic pipeline that maps atomic facts from model-generated text to entity-level annotations, enabling structured hallucination detection at a finer granularity. HALLUENTITY encompasses 18,785 annotated entities, and provides a foundation for evaluating hallucination detection methods with greater interpretability and precision.

Building on HALLUENTITY, we comprehensively evaluate the reliability of token-level uncertainty measurements in detecting hallucinated entities and their potential for localizing hallucinations within the generated text. Our evaluation broadly includes standard uncertainty estimators, such as token-level likelihood (Guerreiro et al., 2023) and entropy scores (Malinin & Gales, 2021), as well as more advanced context-aware approaches that refine uncertainty estimation (Fadeeva et al., 2024; Duan et al., 2024; Zhang et al., 2023b). By aggregating token-level uncertainty to the entity level, we assess whether these methods can accurately distinguish hallucinated entities from factual ones. We experiment with 17 modern LLMs across different model families and capacities. The results reveal that methods solely relying on individual token probabilities (e.g., likelihood and entropy) tend to over-predict hallucinations, making them less reliable. In contrast, context-aware approaches (Duan et al., 2024; Zhang et al., 2023b) demonstrate better overall performance in entity-level hallucination detection. Additionally, model family and size have a limited impact on performance, compared to the choice of uncertainty estimation method, emphasizing the need for improved uncertainty modeling.

Through in-depth qualitative analysis, we further identify relationships between hallucination tendencies and linguistic properties, such as sentence positions and entity types. We found that calibrating uncertainty score with contextual information in the best-performing method (Zhang et al., 2023b) helps reduce over-confidence tendencies in later sentence positions but can unintentionally penalize non-hallucinated content. We also found that some uncertainty scores can frequently assign high uncertainty to informative content like named entities. These observations highlight critical areas for future research, including better modeling of contextual dependencies to maintain balanced precision-recall trade-offs. Our key contributions are summarized as follows:

1. We propose an entity-level hallucination detection dataset, HALLUENTITY, which contains 18,785 annotated entities for ChatGPT-generated biographies.
2. We comprehensively evaluate uncertainty-based hallucination detection approaches across 17 LLMs on our proposed dataset.
3. We conduct an in-depth analysis to identify the strengths and weaknesses of current uncertainty-based approaches, and provide insight to design better uncertainty scores.

## 2 Related Work

**Uncertainty-based hallucination detection methods.** Various approaches have been proposed to detect hallucinated content in LLM generation. Unlike other methods that require external knowledge sources for fact-checking (Gou et al., 2024; Chen et al., 2024a; Min et al., 2023; Huo et al., 2023), uncertainty-based approaches are reference-free and rely only on LLM internal states or behaviors to determine hallucination (Huang et al., 2024; Park et al., 2025). For instance, sampling-based approaches generate multiple responses and measure the diversity in meaning among them (Fomicheva et al., 2020; Kuhn et al., 2023; Lin et al., 2024), while density-based approaches approximate the training data distribution and provide probabilities or unnormalized scores to assess how likely a generated response belongs to the distribution (Yoo et al., 2022; Ren et al., 2023; Vazhentsev et al., 2023).

In this paper, we focus on uncertainty quantification methods that rely on token-level likelihood or entropy (Guerreiro et al., 2023; Malinin & Gales, 2021). Recent works have explored refining likelihood estimation by incorporating semantic relationships or reweighting token importance. For instance, Claim-Conditioned Probability (CCP) (Fadeeva et al., 2024) was introduced to recalculate likelihood according to semantical equivalence; while Zhang et al. (2023b) and Duan et al. (2024) adjust token weights to better convey meaning in uncertainty aggregation. *Although these approaches leverage token-level information, they are typically evaluated at the sentence level, raising questions about their reliability.* To address this, we conduct a comprehensive analysis of entity-level hallucination detection for finer-grained performance insights.

**Fine-grained hallucination detection benchmark.** Most hallucination detection benchmarks are at the sentence or paragraph level. For example, CoQA (Reddy et al., 2019), TriviaQA (Joshi et al., 2017), TruthfulQA (Lin et al., 2022), and HaluEval (Li et al., 2023). These benchmarks classify each generated response as either hallucinated or correct. However, instance-level detection cannot pinpoint specific hallucinated content, which is crucial for correcting misinformation (Cattan et al., 2024). This limitation becomes particularly problematic in long-form text, where a single response often combines supported and unsupported information, making binary quality judgments inadequate (Min et al., 2023).

To address these challenges, recent works have advanced benchmarks for more granular hallucination detection. For example, Min et al. (2023) introduced FACTSCORE, which decomposes LLM-generated text into atomic facts—short sentences conveying a single piece of information—for more precise evaluation. In parallel, Cattan et al. (2024) introduced QASEMCONSISTENCY, decomposing LLM-generated text with QA-SRL, a semantic formalism, to form simple QA pairs, where each QA pair represents one verifiable fact. *However, these methods do not enable entity-level hallucination detection, as they lack explicit entity-level labeling (hallucinated or not) in the original generated text.* Beyond decomposition-based approaches, datasets like HADES (Liu et al., 2022) and CLIFF (Cao & Wang, 2021) create token-level hallucinated content by perturbing human-written text, allowing token-level annotation on the same text. These perturbed hallucinated content, however, could be unrealistic, biased, and overly synthetic due to the limitations of the models they used to perturb words. To bridge this gap, we create a new dataset with entity-level hallucination labels on the same LLMs' generated text. This allows us to evaluate uncertainty-based hallucination detection approaches on a finer-grained level and analyze their reliability.

## 3 HalluEntity: An Entity-Level Hallucination Dataset

### 3.1 Dataset Construction

Curating an entity-level hallucination detection dataset is challenging, requiring annotators to segment sentences into meaningful entities and verify the factual consistency of each entity against reliable sources. This process is time-intensive, requires domain expertise, and is prone to subjectivity (Cao & Wang, 2021). To address these challenges, we first develop a data generation pipeline that maps atomic facts from FActScore (Min et al., 2023) back to the original generated text.

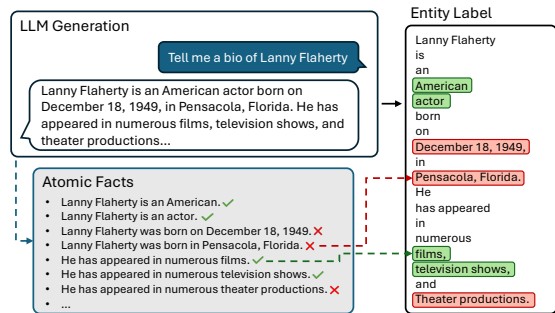

Figure 1: Illustration of our entity-level dataset construction. We form entity-level hallucination labels according to the atomic facts extracted by FACTSCORE.

**Entity segmentation and labeling.** To construct our dataset HALLUENTITY[1], we leverage biographies generated byGPT3.5 (OpenAI, 2023). Each data point consists of a name, a ChatGPT-generated biography, and a list of atomic facts labeled as either `True` or `False`. As illustrated in Figure 1, each atomic fact is a short sentence that conveys a single piece of information. Since atomic facts decompose a sentence into verifiable units, they provide a structured reference for identifying hallucinated entities.

---

[1] HALLUENTITY is publicly released under the MIT license.

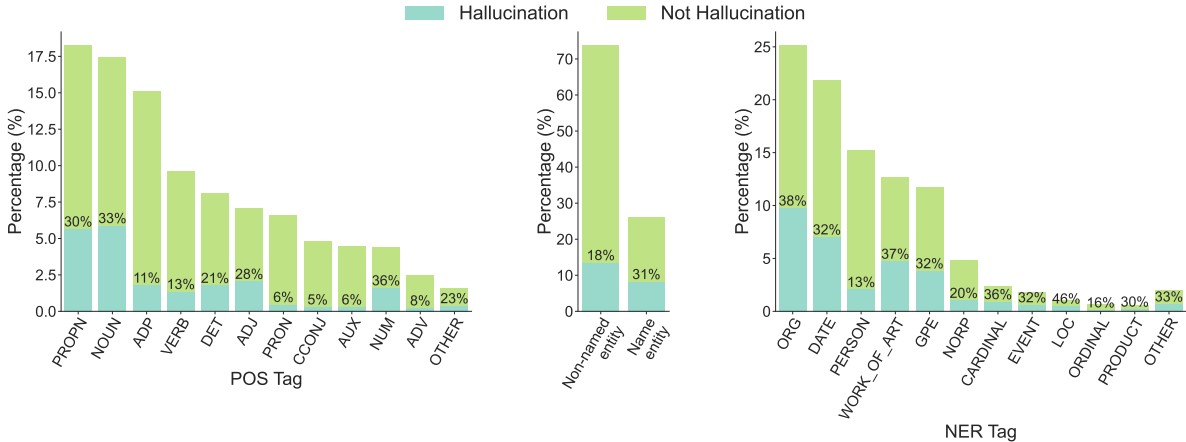

Figure 2: Distribution of POS (left), NER (middle), and breakdown of NER tags (right). Number on each bar indicates the ratio of hallucination for each tag.

To derive entity-level labels, we first segment the original text into meaningful units rather than individual words. For instance, "strategic thinking" is treated as a single entity rather than two separate words. We call such meaningful units *entities*. Given that FACTSCORE decomposes multiple-fact sentences into independent atomic facts, we use these fact-level annotations to label entities. For example, in the sentence "*Lanny Flaherty is an American actor born on December 18, 1949,*" FACTSCORE produces atomic facts:

- "*Lanny Flaherty is an American.*" (True)
- "*Lanny Flaherty was born on December 18, 1949.*"(False)

By aligning these atomic facts with the original text, we label "American" as non-hallucinated and "December 18, 1949" as hallucinated. To scale this process efficiently, we automatically identify and label these entities by instructing GPT-4o (OpenAI, 2024) with a few-shot prompt. Specifically, we manually annotate two examples, each containing an LLM-generated biography, a list of atomic facts, and a corresponding entity-level annotation list. The prompt provides a detailed description of our segmentation method, along with annotated examples. GPT-4o then generates entity labels, which we manually verify and refine to ensure correctness. Further details on the prompt design and annotation process are provided in Appendix A.

## 3.2 Data Analysis

**Data statistics.** HALLUENTITY comprises 157 instances containing a total of 18,785 entities, with 5,452 unique entities. Each entity averages 1.63 words in length. On average, each instance contains 120 entities, with 15% labeled as hallucinated, and 85% as non-hallucinated across the corpus.

**Linguistic feature analysis.** We analyze the relationship between the entity-level hallucination labels and linguistic features, *e.g.*, part-of-speech (POS) and named entities recognition (NER) tags. Specifically, we identify these tags for each word with Spacy (Honnibal & Montani, 2017) and count their occurrence in hallucinated and non-hallucinated entities. The results for each of the part-of-speech (POS) and named entities recognition (NER) tags are shown in Figure 2.

Analysis of POS tags reveals significant patterns in the distribution between hallucinated and non-hallucinated content. Proper nouns (PROPN) constitute the most frequent category with 18.3% occurrences, followed by nouns (NOUN, 17.5%) and adpositions (ADP, 15.1%). Among them, proper nouns and nouns exhibit high hallucination rates of 30.9% and 33.6%, respectively, while adpositions have a lower rate of 11.7%. Moreover, although adjectives (ADJ, 7.1%) and numbers (NUM, 4.4%) are less common, they suffer from a high hallucination rate of 28.9% and 36.2%.

Non-named entities, which comprise 73.8% of total tokens, show a low hallucination rate of 18.2%. In contrast, named entities—despite accounting for only one-third of the tokens—exhibit nearly double the hallucination rate, often exceeding 30%. Among these named entities, person names (PERSON) show the lowest hallucination rate of 13.4%, likely because ChatGPT was prompted to generate biographies for specific individuals.

Beyond POS and NER tagging, hallucination rates vary by position in sentences. The first six words of sentences have a low hallucination rate (9%), but this significantly increases in the middle in the middle (25%) and peaks at the last six words (36%). This comprehensive analysis reveals systematic patterns in hallucination across linguistic features and entity types, providing crucial insights into the reliability of different categories of generated content. In Section 5.3, we see the connections between these linguistic features and the performance of uncertainty-based hallucination detection approaches.

## 4  Uncertainty Scores for Detecting Hallucinated Entities

Given the entity-level hallucination dataset we constructed, a key question arises: ***Can uncertainty scores effectively detect these hallucinated entities***? In this section, we comprehensively explore uncertainty-based methods, all of which measure uncertainty at the token level. These token-level scores can be conveniently aggregated to the entity level, allowing for a systematic evaluation of their effectiveness in identifying hallucinated entities.

**Notations and definitions.**  Let $\mathcal{V}$ be a vocabulary space, $x = (x_1, x_2, \ldots, x_T)$ be a sentence of length $T$ consisting of tokens $x_i \in \mathcal{V}$. The token-level hallucination scores are denoted as $y = (y_1, y_2, \ldots, y_T)$, where $y_i \in \mathbb{R}$. An entity in $x$ is represented as $e_k = (x_i, x_{i+1}, .., x_j)$, where $i$ and $j$ are the start and end indices of the entity's tokens, satisfying $i \leq j \leq T$. For a set of entities $\{e_1, e_2, \ldots, e_K\}$ belong to $x$, where $K$ is the number of entities, their entity-level hallucination labels are defined as $l = (l_1, l_2, \ldots, l_K)$, where $l_k \in \{0, 1\}$ indicates whether $e_k$ is hallucinated. The entity-level scores are computed as $y^e = (y_1^e, y_2^e, \ldots, y_K^e)$, where $y_k^e := \frac{1}{e_{k,1} - e_{k,0} + 1} \sum_{i=e_{k,0}}^{e_{k,1}} y_i$, which aggregates token-level scores to the entity level and $e_{k,0}$ and $e_{k,1}$ are the start and end indices of entity $e_k$. We introduce five methods below to calculate the token-level uncertainty scores.

**Likelihood ([Guerreiro et al., 2023](#)):**  The score is based on the negative log-likelihood of the $i$-th generated token:
$$y_i := -\log p(x_i|x_{<i}).$$

**Entropy ([Malinin & Gales, 2021](#)):**  The score is the entropy of the token probability distribution at position $i$:
$$y_i := -\sum_{v \in \mathcal{V}} p(v|x_{<i}) \log p(v|x_{<i}).$$

**Claim-Conditioned Probability (CCP) ([Fadeeva et al., 2024](#)):**  This method adjusts likelihood based on semantic equivalence using a natural language inference (NLI) model:
$$y_i := -\log \frac{\sum_{k:\texttt{NLI}(x_{<i},x_i^k,x_i)=\text{`e'}} p(x_i^k|x_{<i})}{\sum_{k':\texttt{NLI}(x_{<i},x_i^{k'},x_i)\in\{\text{`e'},\text{`c'}\}} p(x_i^{k'}|x_{<i})},$$

where $x_i^k$ is the $k$-th alternative of the $i$-th generated token, and $\texttt{NLI}$ determines whether concatenating $x_i^k$ with the preceding context entails ('e') or contradicts ('c') the original token. In our experiment, we use the top 10 alternatives and use DeBERTa-base ([He et al., 2021](#)) as the NLI model.

**Shifting Attention to Relevance (SAR) ([Duan et al., 2024](#)):**  This method weights negative log-likelihood by semantic importance:
$$y_i := -\log p(x_i|x_{<i})\widetilde{R_T}(x_i, x),$$

| Approach | AUROC ↑ | AUPRC ↑ | F1$_{\text{Opt}}$ ↑ | Precision$_{\text{Opt}}$↑ | Recall$_{\text{Opt}}$ ↑ |
|---|---|---|---|---|---|
| Likelihood (Guerreiro et al., 2023) | 0.57 | 0.17 | 0.29 | 0.18 | 0.74 |
| Entropy (Malinin & Gales, 2021) | 0.57 | 0.18 | 0.28 | 0.17 | 0.86 |
| CCP (Fadeeva et al., 2024) | 0.57 | 0.25 | 0.26 | 0.15 | **1.00** |
| SAR Duan et al. (2024) | 0.67 | 0.27 | 0.34 | 0.26 | 0.51 |
| Focus (Zhang et al., 2023b) | **0.78** | **0.40** | **0.48** | **0.38** | 0.66 |

Table 1: Performance comparison among five uncertainty scores using Llama3-8B.

where $\widetilde{R_T}$ is $1-$ cosine similarity between the sentence embedding of $x$ and $x \backslash \{x_i\}$. Following Duan et al. (2024), we use SentenceBERT (Reimers & Gurevych, 2019) with RoBERTa-large (Liu et al., 2019) for embedding extraction.

**Focus (Zhang et al., 2023b):** This method refines log-likelihood and entropy using keyword selection, hallucination propagation, and probability correction:

$$y_i := \mathbb{I}(x_i \in \mathcal{K}) \cdot (h_i + \gamma p_i),$$

where $\mathbb{I}(\cdot)$ is an indicator function and $\mathcal{K}$ is keyword set identified by Spacy (Honnibal & Montani, 2017). $h_i$ is the sum of the negative log-likelihood and entropy of $x_i$,

$$h_i := -\log \hat{p}(x_i|x_{<i}) + 2^{-\sum_{v \in \mathcal{V}} \hat{p}(v|x_{<i}) \log_2 \hat{p}(v|x_{<i})}.$$

Here, $\hat{p}(x_i|x_{<i}) = \frac{p(x_i|x_{<i})\text{idf}(x_i)}{\sum_{v \in \mathcal{V}} p(v|x_{<i})\text{idf}(v)}$ is the token probability adjusted by inverse document frequency (IDF), and $p_i$ is the hallucination score propagated from previous tokens,

$$p_i := \sum_{j=0}^{i-1} \frac{\text{att}_{i,j}}{\sum_{k=0}^{i-1} \text{att}_{i,k}} y_j^t,$$

where $\text{att}_{i,j}$ is the attention weight between $x_i$ and $x_j$ after max-pooling for all the layers and attention heads. Following Zhang et al. (2023b), the token IDF is calculated based on 1M documents sampled from RedPajama dataset (Weber et al., 2024), and the hyperparameter $\gamma$ for $p_i$ is set to be 0.9.

Besides these five approaches, we acknowledge that other uncertainty-based hallucination detection approaches exist, such as Semantic Entropy (Kuhn et al., 2023), Verbalized Uncertainty (Kadavath et al., 2022), Lexical Similarity (Fomicheva et al., 2020), EigValLaplacian (Lin et al., 2024), HaloScope (Du et al., 2024), and TSV (Park et al., 2025) However, since these approaches do not produce token-level scores, they are not applicable to our study on detecting hallucinated entites.

# 5 Experiments

## 5.1 Experimental Setup

**Models.** To understand the impact of model family and capacity on entity-level hallucination detection, we experiment with 17 diverse LLMs, including **Llama3**-{8B, 70B} (Llama Team, 2024), **Llama3.1**-8B, **Llama3.2**-3B, **Aquila2**-{7B, 34B} (Zhang et al., 2024), **InternLM2**-{7B, 20B} (Cai et al., 2024), **Qwen2.5**-{7B, 32B} (Qwen, 2025), **Yi**-{9B, 34B} (01.AI, 2024), **phi-2** (Gunasekar et al., 2023), **Mistral**-7B (Jiang et al., 2023), **Mixtral**-8x22B (Jiang et al., 2024), and **Gemma2**-{9B, 27B} (Gemma Team, 2024).

**Evaluation Metrics.** Entity-level hallucination detection can be formulated as a binary classification task. To evaluate performance, we use (1) **AUPRC** and (2) **AUROC**, which assess the relationship between entity-level hallucination labels $l$ and scores $y^e$. These metrics are threshold-agnostic and better suited for comparing uncertainty-based scoring methods. AUPRC captures precision-recall trade-offs, while AUROC

| Model | AUROC | | | | | AUPRC | | | | |
|---|---|---|---|---|---|---|---|---|---|---|
| | Likelihood | Entropy | CCP | SAR | Focus | Likelihood | Entropy | CCP | SAR | Focus |
| [1] meta-llama/Meta-Llama-3-8B | 0.568 | 0.571 | 0.565 | 0.672 | 0.784 | 0.168 | 0.180 | 0.247 | 0.269 | 0.404 |
| [2] meta-llama/Meta-Llama-3-70B | 0.574 | 0.567 | 0.565 | 0.667 | 0.779 | 0.168 | 0.175 | **0.254** | 0.268 | 0.408 |
| [3] meta-llama/Llama-3.1-8B | 0.584 | 0.592 | 0.564 | 0.684 | 0.783 | 0.173 | 0.193 | 0.246 | 0.274 | 0.412 |
| [4] meta-llama/Llama-3.2-3B | 0.577 | 0.591 | 0.564 | 0.685 | 0.772 | 0.170 | 0.191 | 0.235 | 0.269 | 0.368 |
| [5] BAAI/Aquila2-7B | 0.544 | 0.553 | 0.565 | 0.679 | 0.780 | 0.162 | 0.178 | 0.228 | 0.254 | 0.388 |
| [6] BAAI/Aquila2-34B | 0.541 | 0.566 | 0.565 | 0.665 | 0.779 | 0.157 | 0.185 | 0.236 | 0.249 | 0.385 |
| [7] internlm/internlm2-7b | 0.586 | 0.584 | 0.562 | 0.678 | 0.777 | 0.173 | 0.185 | 0.232 | 0.264 | 0.38 |
| [8] internlm/internlm2-20b | 0.579 | 0.573 | 0.561 | 0.674 | 0.77 | 0.171 | 0.179 | 0.233 | 0.267 | 0.349 |
| [9] Qwen/Qwen2.5-7B | 0.557 | 0.571 | 0.558 | 0.675 | 0.767 | 0.164 | 0.185 | 0.220 | 0.255 | 0.346 |
| [10] Qwen/Qwen2.5-32B | 0.561 | 0.569 | 0.559 | 0.674 | 0.768 | 0.166 | 0.183 | 0.226 | 0.258 | 0.347 |
| [11] 01-ai/Yi-9B | 0.541 | 0.549 | 0.56 | 0.663 | 0.776 | 0.159 | 0.173 | 0.231 | 0.237 | 0.375 |
| [12] 01-ai/Yi-34B | 0.543 | 0.543 | 0.557 | 0.653 | 0.769 | 0.156 | 0.165 | 0.229 | 0.233 | 0.349 |
| [13] microsoft/phi-2 | **0.619** | **0.656** | **0.571** | **0.705** | 0.775 | **0.190** | **0.236** | 0.238 | **0.279** | 0.371 |
| [14] mistralai/Mistral-7B-v0.3 | 0.549 | 0.545 | 0.555 | 0.666 | 0.784 | 0.159 | 0.167 | 0.236 | 0.250 | 0.391 |
| [15] mistralai/Mixtral-8x22B-v0.1 | 0.560 | 0.545 | 0.555 | 0.665 | **0.785** | 0.163 | 0.165 | 0.249 | 0.263 | **0.418** |
| [16] google/gemma-2-9b | 0.574 | 0.575 | 0.561 | 0.680 | 0.744 | 0.172 | 0.187 | 0.234 | 0.264 | 0.281 |
| [17] google/gemma-2-27b | 0.576 | 0.566 | 0.557 | 0.673 | 0.780 | 0.174 | 0.177 | 0.232 | 0.263 | 0.397 |

Table 2: AUROC and AUPRC of five uncertainty scores across 17 LLMs.

| Model | $\text{F1}_{\text{Opt}}$ | | | | | $\text{Precison}_{\text{Opt}}$ | | | | | $\text{Recall}_{\text{Opt}}$ | | | | |
|---|---|---|---|---|---|---|---|---|---|---|---|---|---|---|---|
| | Likelihood | Entropy | CCP | SAR | Focus | Likelihood | Entropy | CCP | SAR | Focus | Likelihood | Entropy | CCP | SAR | Focus |
| [1] | 0.290 | 0.278 | 0.261 | 0.344 | 0.484 | 0.180 | 0.166 | 0.150 | 0.261 | 0.384 | 0.742 | 0.860 | **1.000** | 0.505 | 0.658 |
| [2] | 0.291 | 0.282 | 0.261 | 0.335 | 0.469 | 0.182 | 0.168 | 0.150 | **0.274** | 0.339 | 0.736 | 0.870 | **1.000** | 0.432 | 0.757 |
| [3] | 0.296 | 0.286 | 0.261 | 0.351 | 0.483 | 0.188 | 0.180 | 0.150 | 0.268 | 0.362 | 0.696 | 0.688 | **1.000** | 0.505 | 0.724 |
| [4] | 0.294 | 0.285 | 0.261 | 0.349 | 0.477 | 0.177 | 0.182 | 0.150 | 0.256 | 0.348 | 0.855 | 0.662 | **1.000** | 0.548 | 0.758 |
| [5] | 0.283 | 0.275 | 0.261 | 0.346 | 0.484 | 0.169 | 0.162 | 0.150 | 0.248 | 0.365 | **0.875** | 0.916 | **1.000** | 0.570 | 0.719 |
| [6] | 0.283 | 0.277 | 0.261 | 0.329 | 0.484 | 0.171 | 0.168 | 0.15 | 0.222 | 0.361 | 0.838 | 0.786 | **1.000** | **0.632** | 0.732 |
| [7] | 0.304 | 0.286 | 0.261 | 0.348 | 0.475 | 0.192 | 0.174 | 0.150 | 0.260 | 0.385 | 0.736 | 0.797 | **1.000** | 0.525 | 0.620 |
| [8] | 0.293 | 0.281 | 0.261 | 0.348 | 0.467 | 0.181 | 0.169 | 0.150 | 0.272 | 0.332 | 0.771 | 0.820 | **1.000** | 0.481 | **0.783** |
| [9] | 0.291 | 0.277 | 0.261 | 0.342 | 0.475 | 0.175 | 0.166 | 0.150 | 0.237 | 0.347 | 0.857 | 0.833 | **1.000** | 0.613 | 0.753 |
| [10] | 0.293 | 0.280 | 0.261 | 0.341 | 0.482 | 0.179 | 0.166 | 0.150 | 0.251 | 0.356 | 0.817 | 0.883 | **1.000** | 0.535 | 0.749 |
| [11] | 0.285 | 0.276 | 0.261 | 0.332 | 0.482 | 0.172 | 0.162 | 0.150 | 0.233 | 0.362 | 0.843 | **0.936** | **1.000** | 0.580 | 0.721 |
| [12] | 0.289 | 0.279 | 0.261 | 0.323 | 0.478 | 0.175 | 0.166 | 0.150 | 0.227 | 0.353 | 0.831 | 0.880 | **1.000** | 0.559 | 0.739 |
| [13] | **0.315** | **0.323** | **0.266** | **0.361** | 0.477 | **0.195** | **0.215** | **0.261** | 0.254 | 0.348 | 0.810 | 0.650 | 0.270 | 0.622 | 0.760 |
| [14] | 0.289 | 0.276 | 0.261 | 0.333 | **0.489** | 0.176 | 0.162 | 0.150 | 0.237 | **0.386** | 0.811 | 0.933 | **1.000** | 0.561 | 0.666 |
| [15] | 0.293 | 0.280 | 0.261 | 0.334 | 0.471 | 0.177 | 0.167 | 0.150 | 0.256 | 0.345 | 0.836 | 0.866 | **1.000** | 0.477 | 0.743 |
| [16] | 0.294 | 0.282 | 0.261 | 0.346 | 0.476 | 0.180 | 0.168 | 0.150 | 0.258 | 0.344 | 0.809 | 0.856 | **1.000** | 0.526 | 0.770 |
| [17] | 0.296 | 0.282 | 0.261 | 0.345 | 0.473 | 0.182 | 0.166 | 0.150 | 0.250 | 0.349 | 0.800 | 0.916 | **1.000** | 0.556 | 0.732 |

Table 3: $\text{F1}_{\text{Opt}}$, $\text{Precision}_{\text{Opt}}$, and $\text{Recall}_{\text{Opt}}$ of five uncertainty scores across 17 LLMs. Please see Table 2 for models' name.

evaluates true and false positive rates. Unlike AUROC, AUPRC disregards true negatives, emphasizing false positive reduction—a key advantage for hallucination detection, where true negatives often involve less informative entities like prepositions and conjunctions. We complement these metrics by also reporting (3) **$\text{F1}_{\text{Opt}}$**, (4) **$\text{Precision}_{\text{Opt}}$**, and (5) **$\text{Recall}_{\text{Opt}}$**, where $\text{F1}_{\text{Opt}}$ is the optimal F1 score among all possible threshold and $\text{Precision}_{\text{Opt}}$, and $\text{Recall}_{\text{Opt}}$ are corresponding Precision and Recall values.

**Computational resources.** We conducted all experiments on a server equipped with eight Nvidia A100 GPUs. Depending on the model size, each LLM utilized between one to three GPUs. The time required to compute uncertainty scores across the entire dataset varied from 30 seconds to 5 minutes per approach and model, depending on the model size and the complexity of the chosen method.

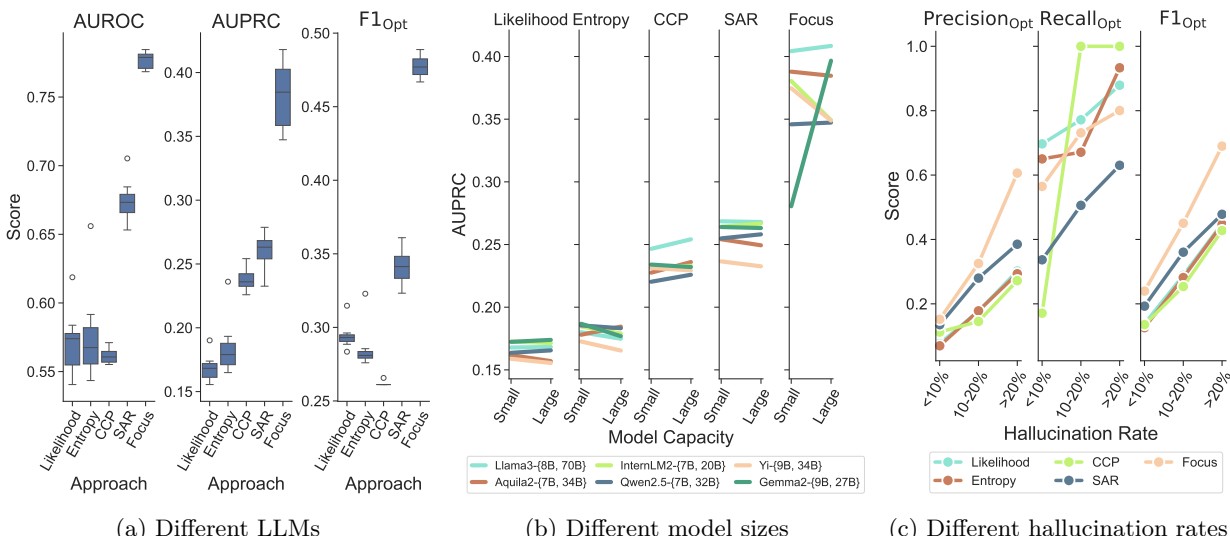

(a) Different LLMs     (b) Different model sizes     (c) Different hallucination rates

Figure 3: For each method, we show performance variation across 17 LLMs (3a). We also show AUPRC scores across LLMs with different capacities (3b), as well as performance on data with different hallucination rates (3c). Note that the model used in Figure 3c is Llama3-8B.

## 5.2 Experimental Results

**How do different uncertainty scores perform on entity-level hallucination detection?** Table 1 presents the evaluation results for five uncertainty-based hallucination detection approaches using Llama3-8B. Likelihood, Entropy, and CCP exhibit low $Precision_{Opt}$ ($\approx$ overall hallucination rate) but achieve high $Recall_{Opt}$. This pattern suggests these methods over-predict hallucinations, making them less suitable for reliable detection. Their focus on individual token probabilities rather than contextual roles likely contributes to this limitation, indicating that *hallucination detection is inherently context-dependent and requires uncertainty scores calibrated with contextual information.*

SAR and Focus, which incorporate context information, show better overall performance. However, their lower $Recall_{Opt}$ indicates that the current methods for modeling context remain suboptimal, failing to capture some hallucinated content. These findings highlight the challenges in entity-level hallucination detection and the need for more advanced approaches that better integrate contextual information while achieving a balanced trade-off between precision and recall.

**How do different LLM families and capacity impact performance?** Table 2, and 3 present the performance of five uncertainty scores across 17 LLMs. The results indicate that `microsoft/phi-2` consistently achieves the highest performance across most scores and evaluation metrics, being the only model that avoids over-predicting hallucinations when using CCP. Additionally, models from Mistral AI (`mistralai/Mistral-7B-v0.3` and `mistralai/Mixtral-8x22B-v0.1`) perform best when using Focus. Notably, phi-2 (2.7B parameters) and Mistral-7B are relatively small models, suggesting that a model's size does not strongly correlate with its ability to detect hallucinations based on token probabilities. While some models outperform others, the performance variations within the same uncertainty score are smaller than those across different scores, as shown in Figure 3a, which summarizes the AUROC, AUPRC, and $F1_{Opt}$ scores vary across model families. This suggests that *the method used to compute uncertainty scores has a more significant impact on performance.*

Figure 3b presents the performance changes across different model sizes within six families: Llama3, Aquila2, InternLM2, Qwen2.5, Yi, and Gemma2, each comprising two size variants. The results reveal that, in most cases, using a larger model does not significantly enhance performance. The only exception is using Gemma on Focus, where the AUROC score improves by 0.12 between the 27B and 9B versions. Performance improvements for other model families and approaches remain marginal, typically below 0.01. These findings

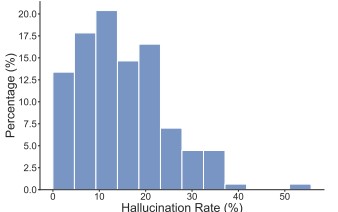

Figure 4: Distribution of entity-level hallucination rate. Most instances in our dataset have a hallucination rate $< 25\%$.

| | Low | Medium | High |
|---|---|---|---|
| Hallucination Rate | $<10\%$ | 10-20% | $>20\%$ |
| # of Instance | 54 | 59 | 44 |

Table 4: Statistics of data grouped according to the hallucination rate. Each group has a similar amount of data.

suggest that *a larger LLM may not reflect its better capability of determining hallucination on its token probabilities.*

**How does performance vary across different hallucination levels?** We categorize HALLUENTITY into three groups based on the hallucination rate—the proportion of hallucinated entities in each generation. Figure 4 shows the distribution of hallucination rate. The results indicate that most biographies generated by ChatGPT have a hallucination rate below 25%. Additionally, generations with hallucination rates below 10% and those between 10% to 20% occur at similar frequencies. Based on this observation, we categorize the data into three groups ($< 10\%$, 10-20%, $> 20\%$) to examine how hallucination rates impact detection performance. Table 4 summarizes the group statistics, showing that each group contains a similar amount of data. Figure 3c shows the $\text{Precision}_{\text{Opt}}$, $\text{Recall}_{\text{Opt}}$, and $\text{F1}_{\text{Opt}}$ scores across three groups. The results reveal that all methods struggle to detect hallucinated content when the hallucination rate is low, with $\text{F1}_{\text{Opt}}$ scores around 0.2. Entropy and CCP exhibit a steep increase in $\text{Recall}_{\text{Opt}}$ compared to $\text{Precision}_{\text{Opt}}$ as the hallucination rate increases, suggesting their tendency to over-predict hallucinations, particularly in a high-hallucination scenario. In contrast, Focus achieves a small difference between the $\text{Recall}_{\text{Opt}}$ and $\text{Precision}_{\text{Opt}}$ when the hallucination rate is high, demonstrating its ability to balance precision-recall trade-offs while also highlighting the challenge of detecting sparse hallucination.

### 5.3 In-depth Analysis

To better understand the strengths and limitations of uncertainty scores for detecting hallucinated entities, we analyze cases where (1) all scores failed or misidentified hallucinations, and (2) scores varied in performance. We classify entities using thresholds for $\text{F1}_{\text{Opt}}$ and categorize false positives/negatives by POS, NER tags, and sentence positions (first, middle, or last six words). We then identify tags and positions where approaches excel or falter, visualizing samples with color-coded uncertainty scores to uncover patterns behind detection discrepancies (See Table 5). Figure 5 shows the FPR and FNR across NER tags and sentence positions, and Figure 6 shows the FPR/FNR across POS tags. Our analysis focuses on Likelihood, SAR, and Focus, as SAR and Focus demonstrated the most effective performance in Section 5.2, and Likelihood serves as a straightforward baseline for comparison. Note that Entropy has a similar trend of FPR/FNR across POS and NER tags compared to Likelihood. Thus, we primarily report the result of Likelihood for simplicity.

**SAR under-predicts hallucinations due to unreliable token importance weighting.** The left and middle plots of Figure 5 show that SAR has the lowest FPR but the highest FNR across most tags, particularly for named entities, indicating a tendency to under-predict hallucinations. Consistent with Figure 5, Figure 6 shows that SAR demonstrates lower FPR but higher FNR across POS tags. This occurs because SAR weights token importance based on sentence similarity without the token, which often remains unchanged even if the token is informative. The first case in Table 5 illustrates this: SAR assigns lighter shades to entities like the second "Santa Cruz" since removing either "Santa" or "Cruz" barely affects sentence similarity, despite the term's informativeness.

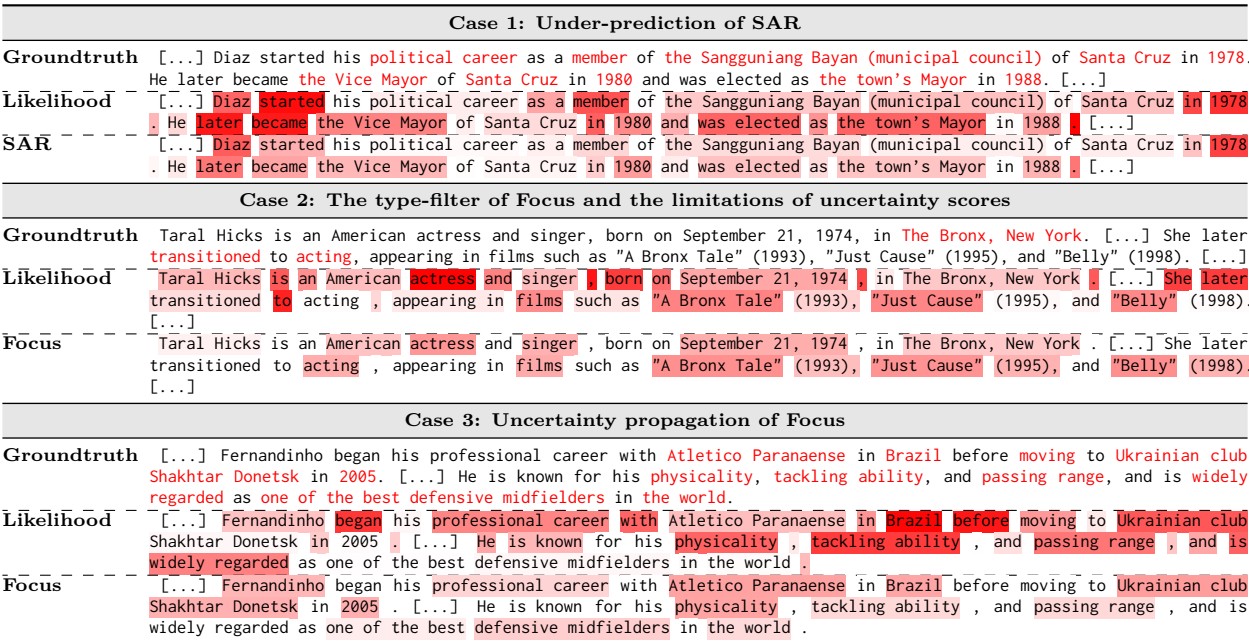

Table 5: We sampled 3 instances from our dataset to demonstrate the differences across uncertainty scores. For label, entities colored in red indicate hallucination. For uncertainty scores, entities boxed in red with different tints represent the degree of uncertainty. A lighter (darker) box indicates a lower (higher) uncertainty.

**The type-filter of Focus reduces FNR on name entities but sheds light on a bigger limitation of uncertainty-based hallucination score.** The left and middle plots of Figure 5 reveal that Focus performs differently for named and non-named entities. It achieves a low FNR but high FPR for named entities, and the opposite for non-named ones. This is because Focus filters for named entities based on POS and NER tags. While promising—since named entities often hallucinate (as shown in Figure 2)—its high FPR suggests that its base score (the sum of Likelihood and Entropy) poorly distinguishes hallucinations, frequently assigning high uncertainty to named entities. Such high uncertainty on correct named entities often arises when the correct information can be expressed in multiple valid ways. For example, in the generation, "*She later transitioned to acting, appearing in films such as 'A Bronx Tale' (1993), 'Just Cause' (1995), and 'Belly' (1998),*" the order of the listed films can vary without affecting factual accuracy. In this scenario, there are several plausible next tokens following "appearing in films such as," each with a relatively low predicted probability, resulting in high entropy despite the content being correct.

Similarly, in Figure 6, Focus shows varying patterns depending on the POS tags. For proper nouns, nouns, and numbers—tags often associated with named entities—Focus has a higher FPR and lower FNR. However, for verbs, auxiliaries, and adverbs, Focus exhibits a lower FPR but a higher FNR. This highlights a limitation of Focus: by concentrating primarily on named entities, it tends to overlook hallucinations in other types of tokens. The 2nd case in Table 5 illustrates this: Focus ignores function words like "is" and "to," reducing FPR, but indiscriminately highlights named entities like "American" and "A Bronx Tale," even when accurate.

**Uncertainty propagation of Focus alleviates the over-confidence nature of LLMs.** The right plots in Figure 5 show that LLMs are less confident when generating the first few words of a sentence and become over-confident as generation progresses, as indicated by a decrease in FPR and an increase in FNR for Likelihood. This contrasts with the typical distribution of hallucinations, which occur mostly in the middle and end of sentences (Section 3.2). Focus addresses this by propagating uncertainty scores based on attention, leading to a decrease in FNR over positions. However, its FPR increase over positions suggests that using attention scores to propagate uncertainty may wrongly penalize entities that are not over-confident. The 3rd case in Table 5 illustrates this: Likelihood assigns higher uncertainty to early words (*e.g.*, "Fernandinho

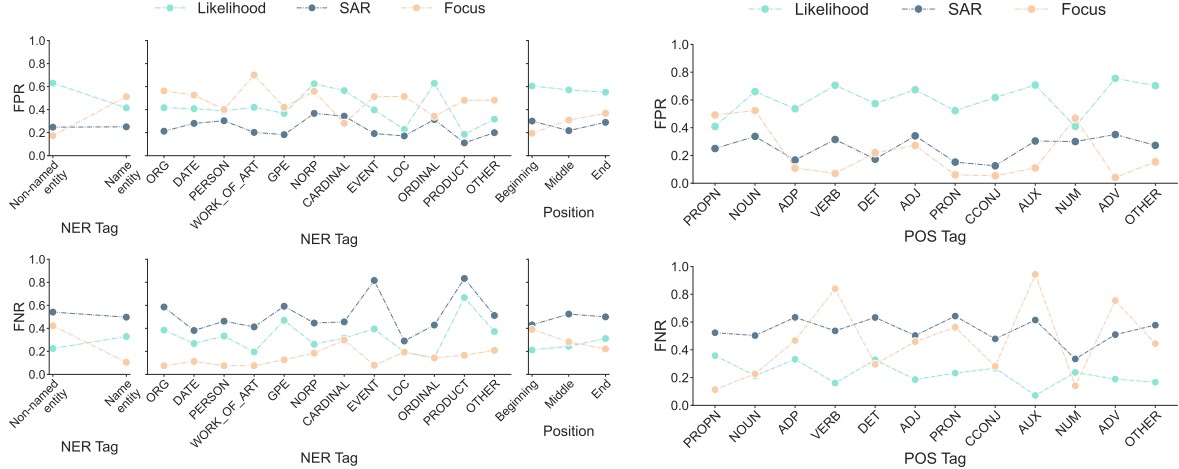

Figure 5: FPR/FNR of uncertainty scores across NER tags (left and middle) and sentence positions (right).

Figure 6: FPR/FNR of each uncertainty score across POS tags.

began") and lower scores to later words (*e.g.*, "Shakhtar Donetsk in 2005"), while Focus detects hallucinations at sentence ends by linking them to prior hallucinated content (*e.g.*, "Ukrainian club").

# 6 Discussion

Throughout this study, we identify the limitations and strengths of current uncertainty-based hallucination detection approaches. In this section, we discuss two directions to improve the performance of uncertainty-based hallucination detection.

**Uncertainty score calibration.** In Section 5.3, we show how simple calibrations (type-filter and uncertainty propagation) help improve the performance of Focus. These calibrations were invented through the linguistic analysis of LLM-generated corpus, indicating the relationship between the tendency of hallucination and linguistic properties. Based on this finding, we recommend exploring more linguistic properties that can help determine the importance of generating content and the tendency of hallucination. This exploration would not only improve the performance of hallucination detection, but also help mitigate hallucination during generation.

**Utilizing information beyond token probabilities to estimate uncertainty.** One major limitation of uncertainty-based hallucination detection is its over-prediction nature. As shown in Section 5.2, all five approaches perform poorly when hallucinations are sparse. In Section 5.3, we further show that such over-prediction frequently happens on informative content, such as name entities. This suggests that token probabilities are not well separated between hallucinated and non-hallucinated content, and using uncertainty scores like Likelihood or Entropy to serve as the base score of hallucination detection is not reliable. Hence, we recommend investigating more sophisticated uncertainty estimation or integrating probing techniques that utilize other information from LLM's internal states to increase the reliability of hallucination detection.

**Conclusion and future work.** In this work, we comprehensively explore the promise of entity-level hallucination detection by curating HALLUENTITY, a dataset tailored for fine-grained understanding and introducing evaluation metrics for the task. We benchmark five uncertainty-based approaches, finding that they struggle to localize hallucinated content, raising concerns about their reliability. Our qualitative analysis highlights their strengths and weaknesses and suggests two directions for improvement. Future work should explore more sophisticated techniques for incorporating context-aware uncertainty estimation and develop methods that adaptively propagate uncertainty across sentence positions to enhance hallucination localization.

## Limitations

**Dataset constructions.**  Since curating a hallucination detection dataset from scratch is challenging, especially when annotating hallucinations at the entity level, we build our dataset upon the FActScore dataset to reduce cost and enhance quality. However, this means our dataset would inherit limitations from it. For example, the FActScore dataset focuses only on biography generation but omits other common tasks such as QA and data analysis. In addition, since the FActScore dataset verify claims with Wikipedia, which does not contain all the information to verify a biography, some factual claims may be missclassified as "non-support." During our annotation process, we observe cases where the FActScore label was "non-support" while GPT-4o annotated the corresponding entity as "True." For these conflict cases, we manually verified them through Google search and updated the labels if needed.

**Evaluation of hallucination detection approaches.**  In this paper, we focus on evaluating uncertainty-based hallucination detection approaches, where the uncertainty scores are estimated by token probabilities. For other types of uncertainty estimation that measure the diversity across samples, such as Semantic Entropy, since they estimate uncertainty at the sample level and do not output scores for each token or entity, they can not be evaluated on HALLUENTITY. Although this incompatibility limits the usage of HALLUENTITY, it also shows the limitation of sample-based approaches—they are hard to pinpoint hallucinated content.

## Ethical Statement

This research addresses the critical challenge of hallucination detection in LLMs to enhance their safe and responsible use across high-stakes domains. By exploring entity-level hallucination detection and evaluating uncertainty-based methods, we aim to improve the precision and reliability of identifying factual inaccuracies in generated content. HALLUENTITY and evaluation metrics are intended solely for research purposes, ensuring no sensitive or personal information is included. We acknowledge the limitations of current approaches and advocate for continued improvements to promote transparency, accuracy, and responsible AI development.

## Acknowledgement

Research is supported in part by the AFOSR Young Investigator Program under award number FA9550-23-1-0184, National Science Foundation (NSF) Award No. IIS-2237037 and IIS-2331669, Office of Naval Research under grant number N00014-23-1-2643, Schmidt Sciences Foundation, Open Philanthropy, Alfred P. Sloan Fellowship, and gifts from Google and Amazon.

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

# A  Details of Dataset Construction

**Selection of data.**  From the FACTSCORE dataset, we select the set of biographies generated by ChatGPT to construct our entity-level hallucination detection dataset. The set of ChatGPT-generated biographies contains 183 samples. We filter out those that ChatGPT refuses to answer and end up with 157 instances.

**Data labeling process.**  For each sample, FACTSCORE provides a list of atomic facts—short sentences conveying single pieces of information. These facts are labeled as `Supported`, `Not-supported`, or `Irrelevant`, where `Irrelevant` means the fact is unrelated to the prompt (*i.e.* a person's name), and `Supported` and `Not-supported` indicate whether the fact is supported by the person's Wikipedia page. Since only 8.3% of facts are labeled as `Irrelevant`, and most are related to `Not-supported` facts, we simplify the entity-level labeling process by merging both as `False`, treating only `Supported` facts as `True`.

To assign entity-level labels, we first tokenize the biography into individual words. We then use the list of atomic facts to group words into meaningful units (entities) and assign labels based on fact types. Specifically, for atomic facts that share a similar sentence structure (*e.g.*, *"He was born on Mach 9, 1941."* (`True`) and *"He was born in Ramos Mejia."* (`False`)), we label differing entities first—assigning `True` to "Mach 9, 1941" and `False` to "Ramos Mejia.". For those entities that are the same across atomic facts (*e.g.*, "was born") or are neutral (*e.g.*, "he," "in," and "on"), we label them as `True`. In cases where no similar atomic fact exists, we identify the most informative entities in the sentence, label them based on the atomic fact, and treat the remaining entities as `True`. Note that we assume the pronouns in generations are always correct, as in biography generation, they usually refer to the person specified in the prompt. When a claim is non-supported, we consider other information related to the pronouns as a hallucination.

**GPT-4o prompt for data labeling.**  To scale the labeling process, we use GPT-4o to automatically identify and label entities with a few-shot prompt, as shown in Table 6. The system prompt includes detailed instructions on the labeling process, along with two manually created examples. In the user prompt, we maintain the same structured format used in the examples, inputting the biography and the corresponding list of atomic facts.

**Data quality assessment.**  We performed three run annotations for a random subset of samples. We then compute the Jaccard similarity across the three entity sets. The similarity score is 0.813, suggesting a reasonably high similarity. We further compute Fleiss's Kappa inner-annotator agreement on entities' label (True/False) where the entities are the same across three runs. The Fleiss's Kappa across the three runs is 0.984, suggesting a near-perfect agreement.

In addition to multiple-run annotation, we also conduct human verification to assess the annotation quality. The verification was conducted by the authors of the paper, who are well familiar with hallucination detection and LLM. To verify the GPT-4o annotation, we provided the generated biography and original FActScore annotation as references. The verification includes two parts: 1) whether a sentence is segmented into reasonable entities. For example, "December 18, 1949" should be one entity instead of three. 2) whether the label of an entity aligns with the original FActScore annotation. For example, given an original annotation, "Lanny Flaherty was born on December 18, 1949 (non-support)," the entity, "December 18, 1949," should be labeled as False.

The authors verified the whole dataset independently. When a discrepancy between the verification results exists, the annotators would discuss it to reach a final consensus. The Jaccard similarity between the GPT-4o annotation and the annotation after verification is 0.945, and the Cohen's Kappa inner-annotator agreement on entities' label (True/False) where the entities are the same before/after verification is 0.940. This result suggests that the GPT-4o annotation is reliable.

# B  Proxy LLM Setting

Our experiments are conducted under the proxy LLM setting, *i.e.*, using different LLMs for hallucination detection and for response generation. This setting assumes the generator and detector LLMs are sufficiently

---

**System prompt**

You are a helpful and precise assistant for segmenting and labeling sentences. We would like to request your help on curating a dataset for entity-level hallucination detection.

We will give you a machine generated biography and a list of checked facts about the biography. Each fact consists of a sentence and a label (True/False). Please do the following process. First, breaking down the biography into words. Second, by referring to the provided list of facts, merging some broken down words in the previous step to form meaningful entities. For example, "strategic thinking" should be one entity instead of two. Third, according to the labels in the list of facts, labeling each entity as True or False. Specifically, for facts that share a similar sentence structure (*e.g.*, *"He was born on Mach 9, 1941."* (True) and *"He was born in Ramos Mejia."* (False)), please first assign labels to entities that differ across atomic facts. For example, first labeling "Mach 9, 1941" (True) and "Ramos Mejia" (False) in the above case. For those entities that are the same across atomic facts (*e.g.*, "was born") or are neutral (*e.g.*, "he," "in," and "on"), please label them as True. For the cases that there is no atomic fact that shares the same sentence structure, please identify the most informative entities in the sentence and label them with the same label as the atomic fact while treating the rest of the entities as True. In the end, output the entities and labels in the following format:
- Entity 1 (Label 1)
- Entity 2 (Label 2)
- ...
- Entity N (Label N)
Here are two examples:

**[Example 1]**
[The start of the biography]
Marianne McAndrew is an American actress and singer, born on November 21, 1942, in Cleveland, Ohio. She began her acting career in the late 1960s, appearing in various television shows and films.
[The end of the biography]

[The start of the list of checked facts]
[Marianne McAndrew is an American. (False); Marianne McAndrew is an actress. (True); Marianne McAndrew is a singer. (False); Marianne McAndrew was born on November 21, 1942. (False); Marianne McAndrew was born in Cleveland, Ohio. (False); She began her acting career in the late 1960s. (True); She has appeared in various television shows. (True); She has appeared in various films. (True)]
[The end of the list of checked facts]

[The start of the ideal output]
[Marianne McAndrew (True); is (True); an (True); American (False); actress (True); and (True); singer (False); , (True); born (True); on (True); November 21, 1942 (False); , (True); in (True); Cleveland, Ohio (False); . (True); She (True); began (True); her (True); acting career (True); in (True); the late 1960s (True); , (True); appearing (True); in (True); various (True); television shows (True); and (True); films (True); . (True)]
[The end of the ideal output]

**[Example 2]**
[The start of the biography]
Doug Sheehan is an American actor who was born on April 27, 1949, in Santa Monica, California. He is best known for his roles in soap operas, including his portrayal of Joe Kelly on "General Hospital" and Ben Gibson on "Knots Landing."
[The end of the biography]

[The start of the list of checked facts]
[Doug Sheehan is an American. (True); Doug Sheehan is an actor. (True); Doug Sheehan was born on April 27, 1949. (True); Doug Sheehan was born in Santa Monica, California. (False); He is best known for his roles in soap operas. (True); He portrayed Joe Kelly. (True); Joe Kelly was in General Hospital. (True); General Hospital is a soap opera. (True); He portrayed Ben Gibson. (True); Ben Gibson was in Knots Landing. (True); Knots Landing is a soap opera. (True)]
[The end of the list of checked facts]

[The start of the ideal output]
[Doug Sheehan (True); is (True); an (True); American (True); actor (True); who (True); was born (True); on (True); April 27, 1949 (True); in (True); Santa Monica, California (False); . (True); He (True); is (True); best known (True); for (True); his roles in soap operas (True); , (True); including (True); in (True); his portrayal (True); of (True); Joe Kelly (True); on (True); "General Hospital" (True); and (True); Ben Gibson (True); on (True); "Knots Landing." (True)]
[The end of the ideal output]

**User prompt**

[The start of the biography]
{BIOGRAPHY}
[The end of the biography]

[The start of the list of checked facts]
{LIST OF CHECKED FACTS}
[The end of the list of checked facts]

Table 6: GPT-4o prompt for labeling hallucinated entities.

similar in terms of training data and architecture. These assumptions are reasonable in practice. For architectural similarity, most large language models today adopt the Transformer-based decoder-only architecture, following the GPT paradigm. This includes both the target models we evaluate and the proxy models used for detection. Therefore, the structural inductive biases are highly aligned between the generator and the

| Proxy model / Target model | Llama3-8B | Llama3.1-8B | InternLM2-7B | Qwen2.5-7B | Phi2 | Mistral-7B | Gemma2-9B |
|---|---|---|---|---|---|---|---|
| Llama3-8B | 1.000 | 0.978 | 0.896 | 0.896 | 0.854 | 0.916 | 0.925 |
| Mistral-7B | 0.908 | 0.907 | 0.892 | 0.865 | 0.829 | 1.000 | 0.923 |
| Gemma2-9B | 0.931 | 0.930 | 0.897 | 0.897 | 0.849 | 0.929 | 1.000 |

Table 7: Pearson's correlation of log-likelihood across models.

| Proxy model / Target model | Llama3-8B | Llama3.1-8B | InternLM2-7B | Qwen2.5-7B | Phi2 | Mistral-7B | Gemma2-9B |
|---|---|---|---|---|---|---|---|
| Llama3-8B | 1.000 | 0.982 | 0.906 | 0.893 | 0.839 | 0.922 | 0.925 |
| Mistral-7B | 0.923 | 0.925 | 0.891 | 0.884 | 0.800 | 1.000 | 0.932 |
| Gemma2-9B | 0.941 | 0.938 | 0.901 | 0.911 | 0.847 | 0.941 | 1.000 |

Table 8: Pearson's correlation of entropy across models.

proxy. In addition, for training data similarity, the large-scale pretraining corpora used by many open-source LLMs show strong overlap, often including subsets of Common Crawl, Wikipedia, GitHub, and other widely shared internet-sourced datasets. While we may not have access to the exact training sets of closed models, this convergence in training data sources leads to similar token-level uncertainty patterns, especially on general-domain QA tasks.

**Empirical validation.** To validate this hypothesis more rigorously, we compute Pearson's correlation between uncertainty scores produced by the target model and those produced by a proxy model, across multiple uncertainty scores (*e.g.*, Likelihood and Entropy). Specifically, we generate biographies with an open-sourced LLM (*e.g.*, Llama3-8B) and compute uncertainty scores using the same model as well as other proxy models. We observe consistently high Pearson's correlations in Table 7 and 8. This confirms that proxy uncertainty is a faithful surrogate.

