# OpenReview forum: "HalluEntity: Benchmarking and Understanding Entity-Level Hallucination Detection"
_TMLR — Accepted by TMLR_

### Review · Reviewer_3S6k · 2025-04-07

**Summary Of Contributions:**

The authors present a study on whether token-level uncertainty scores can detect hallucinated entities.
For this purpose, they create a dataset, HalluEntity, that is obtained by asking ChatGPT to write biographies about different people, which are split into verifiable facts that are manually verified.
The authors then propose a first analysis on the hallucination rates per PoS / NER tag.
Afterwards, they test 5 token-level uncertainty quantification method (likelihood, entropy, claim-conditioned probability, shifting attention to relevance, and focus), across 17 different LLMs.
They follow-up by detailed analyses on the effect of false-positive and false-negative rates per NER tag and sentence position.

**Audience:**

Yes

**Broader Impact Concerns:**

Any concerns are addressed in the Ethical Statement.

**Claims And Evidence:**

Yes

**Requested Changes:**

What follows is an unordered list of requested changes / additional questions:

* In the second paragraph on page 2, \citep should be used for the citation.
* The vertical spacing underneath FIgure 1 seems to be a bit off.
* I personally found the ratio of hallucinations in Figure 2 with the secondary y-axis a bit confusing - maybe this could be replaced by a percentage value inside the bars of the stacked bar plot?
* I would like to ask the authors to specify the exact version of ChatGPT they used to generate biographies.
* The notation in section 4 appears a little cluttered; First of all, defining a sentence x with tokens v_i seems a bit unintuitive, but importantly clashes with the notation of the uncertainty metrics (e.g. likelihood should be -log p(v_i|v_{<i}) then). Secondly, it also feels unintuitive that a sentence x could be rewritten as a series of entities, rather I would define a set of entities belonging to a sentence x. Token-level halucination scores have a superscript t supposedly to distinguish them from the entitiy level hallucination scores. But t is easy to confuse for the timestep since sentences have length T. It might be easier to simply drop the superscript t for $y_t$.
* For SAR, it looks the like the tilde for the relevance weight is over $R_T$ instead of just $R$.
* For Focus,  the indicator function can be inferred from the context, but is notation is not introduced.

**Strengths And Weaknesses:**

Strengths
-----------

* The study is thorough in its coverage of token-level uncertainty measures and language models.
* The paper presents a lot of very interesting, additional experiments. From the composition of their dataset, to performances as grouped by uncertainty method, model size, and hallucination rate. The FPR and FNR analyses of the results for SAR, likelihood and Focus are also intriguing.

Weaknesses
---------------

* For me, there is one glaring issue with this paper: For other types of hallucination detection, it makes sense to evaluate an external detector applied to some target model (in this case ChatGPT). The reason why uncertainty *might* be a good indicator for hallucinations is that it should be high when the model struggles to process ambiguous data, or generates plausible but unfounded claims that do not have any grounding in the training data. However, this study looks at the uncertainty of an external model to detect hallucinations in a target model. I can see that sometimes they might coincide, but especially for smaller models from completely different families this seems very questionable. I feel this study unfortunately does not possess much cogency unless it applies similar methods to the token-level uncertainties of the target model, and then potentially shows a relationship between uncertainties of the target and external models. I think that releasing the HalluEntity dataset together with token probability distributions and hidden representations of the target model would greatly improve their contribution.
* Other minor weaknesses are elaborated on in the next section.

---

> ### Author Response · Authors · 2025-07-02
>
> We sincerely appreciate your feedback and insightful comments! We address the questions and comments below in detail.
>
> > Clarification on proxy LLM
>
> Thank you for the thoughtful critique. We agree that analyzing hallucinations based on the internal uncertainty of the generation model (e.g., ChatGPT) would be ideal. However, in practice, many deployed LLMs are closed-source and do not expose token-level uncertainties or hidden states. Our study is thus motivated by a realistic and practically relevant scenario: can external models detect hallucinated entities in outputs from any systems (including black-box ones)? This is especially important for building downstream safety tools that must operate without internal access to the generator.
>
> While we acknowledge the potential limitations of model mismatch, our extensive experiments across 17 LLMs—including varying sizes and families—reveal meaningful patterns and demonstrate that external uncertainty estimates can still provide useful signals. Moreover, prior work [1, 2] has shown promising results using proxy LLMs to detect sentence-level hallucinations via uncertainty signals. Our work extends this line of research to the more fine-grained entity level. We agree that releasing token-level outputs from the target model would be valuable, and we are exploring ways to expand HalluEntity in that direction where feasible. We view this work as a first step toward practical entity-level hallucination detection under realistic constraints, and hope it encourages further research on bridging target and detector models. In our revised manuscript, we have additionally discussed this issue in the limitation section.
>
> > ChatGPT version for biography generation
>
> The responses are generated by GPT3.5.
>
> > Other requested changes
>
> We also thank you for the suggestions for the presentation and notations, which we've updated on the manuscript.
>
> [1]: Zhang, Tianhang et al., Enhancing Uncertainty-Based Hallucination Detection with Stronger Focus. 2023
> [2]: Duan, Jinhao et al., Shifting Attention to Relevance: Towards the Predictive Uncertainty Quantification of Free-Form Large Language Models. 2024

---

> > ### Comment · Reviewer_3S6k · 2025-07-03
> > **Response to Author Comment**
> >
> > I thank the authors for their reply to my review for responding to my concerns.
> >
> > I would like to point out that [2] does not evaluate uncertainty estimates as hallucination detection signals in another model, which makes this claim misleading. [1] indeed tests a similar scenario like in the paper.
> >
> > One aspect of the experimental results presented in the paper make me slightly worried regarding, especially reagrding my main point of critique:
> > Most model  and uncertainty metric scores in Tables 2 + 3 only hover marginally over a random AUROC of 0.5 and the expected AUPRC given the class distribution (~ 15 % hallucination).
> > The ones that do (like microsoft/phi2 and the SAR and FOCUS metrics) report that either precision and / or recall suffer for most settings, which single but inconsistent exceptions. Since hallucination detection in practice would be employed with a fixed tuned decision threshold, this would suggest either a stark amount of false positives or missed hallucinations, i.e. false negatives (I also believe that the second result column Table 12 also should read AUPR, which I missed during my first review).
> >
> > I could imagine that a secondary model could be used to detect the target model's hallucination through uncertainty if they are sufficiently similar in terms of training data and architecture. However, I don't find the presented results (despite the great presentation and unmistaken amount of effort by the authors) not yet fully convincing.
> >
> > I think the results would profit from the inclusion of some simple baselines (such as a majority baseline and random baseline based on the label distribution) and some non-uncertainty based hallucination detection baselines, as also done in the cited [1] with SelfCheckGPT, or other metods.
> >
> > Further, I think the (potential) danger of this approach is that under the above assumption ("uncertainty-based hallucination detection can work with a proxy model if they are sufficiently similar") , we would assume the detector to degrade on OOD data, where the target and detection model would divergence more in their behavior. I would like to see an experiment in this regard that could demonstrate a more gentle degradation of performance compared to baselines that are not based on uncertainty.

---

> > > ### Author Response · Authors · 2025-07-07
> > > **Response to follow up comments (part 1)**
> > >
> > > We sincerely appreciate your follow up comments. We address the them below in detail.
> > >
> > > > Mistake in references
> > >
> > > We apologize for the mistake made in the response. We will remove the SAR paper from the citation.
> > >
> > > > Clarification on performance
> > >
> > > We would like to clarify that the primary goal of this paper is not to claim that uncertainty alone solves hallucination detection at high AUROC. Rather, our aim is to highlight a previously under-explored and inherently difficult problem: hallucination detection at the entity level without access to the generating model. We believe this setting is both practically important and scientifically under-studied. By systematically benchmarking a diverse set of uncertainty metrics across a range of LMs and tasks, we hope to:
> > >
> > > (1) establish baseline understandings of the potential and limitations of uncertainty signals,
> > >
> > > (2) call attention to the challenging nature of model-free hallucination detection, and
> > >
> > > (3) stimulate future work on more advanced detection techniques, potentially going beyond uncertainty-based approaches with content-based verification or consistency checks.
> > >
> > > We believe this honest assessment of the limits of current methods, rather than over-claiming performance, is a critical step toward building entity-level hallucination detectors. Finally, we thank the reviewer for catching the error in Table 12 — the second column indeed reflects AUPRC, and we will correct the labeling in the final version.
> > >
> > > > Precision/Recall tradeoff
> > >
> > > As noted, no single threshold gives perfect precision and recall. This is a well-known challenge in uncertainty quantification and binary classification, especially with imbalanced datasets like ours. This precision–recall tradeoff is not a failure of uncertainty modeling per se, but a reflection of the complex, fine-grained nature of the hallucination detection task.
> > >
> > > To help clarify this, our analysis prioritizes ranking quality (via AUROC and AUPRC), which is threshold-agnostic and better suited for comparing uncertainty-based scoring methods. In real-world deployment, thresholds can be tuned based on the specific application’s tolerance for false positives vs. false negatives (e.g., conservative filtering vs. high-recall flagging). Nevertheless, we understand the reviewer’s concern and agree that improving thresholded detection remains a key challenge.
> > >
> > > > _"I could imagine that a secondary model could be used to detect the target model's hallucination through uncertainty if they are sufficiently similar in terms of training data and architecture._"
> > >
> > > We would like to clarify this concern in three aspects:
> > > - **Architectural similarity**: Indeed, most large language models today adopt the Transformer-based decoder-only architecture, following the GPT paradigm. This includes both the target models we evaluate and the proxy models used for detection. Therefore, the structural inductive biases are highly aligned between the generator and the proxy.
> > >
> > > - **Training data similarity**: The large-scale pretraining corpora used by many open-source LLMs also show strong overlap, often including subsets of Common Crawl, Wikipedia, GitHub, and other widely shared internet-sourced datasets. While we may not have access to the exact training sets of closed models, this convergence in training data sources leads to similar token-level uncertainty patterns, especially on general-domain QA tasks.
> > >
> > > - **Empirical verification of proxy-target alignment**: To validate this hypothesis more rigorously, we compute the pearson correlation between uncertainty scores produced by the target model and those produced by a proxy model, across multiple uncertainty scores (e.g., Likelihood and Entropy). Specifically, we generate biographies with an open-sourced LLM (e.g., Llama3-8B) and compute uncertainty scores using the same model as well as other proxy models. **We observe consistently high pearson correlations in below tables**. This confirms that proxy uncertainty is a faithful surrogate.
> > >
> > > #### Likelihood
> > >
> > > |Target model\Proxy model|Llama3-8B|Llama3.1-8B|InternLM2-7B|Qwen2.5-7B|Phi2|Mistral-7B|Gemma2-9B|
> > > |---|---|---|---|---|---|---|---
> > > |Llama3-8B|1.000|0.978|0.896|0.896|0.854|0.916|0.925|
> > > |Mistral-7B|0.908|0.907|0.892|0.865|0.829|1.000|0.923|
> > > |Gemma2-9B|0.931|0.930|0.897|0.897|0.849|0.929|1.000|
> > >
> > > #### Entropy
> > >
> > > |Target model\Proxy model|Llama3-8B|Llama3.1-8B|InternLM2-7B|Qwen2.5-7B|Phi2|Mistral-7B|Gemma2-9B|
> > > |---|---|---|---|---|---|---|---
> > > |Llama3-8B|1.000|0.982|0.906|0.893|0.839|0.922|0.925|
> > > |Mistral-7B|0.923|0.925|0.891|0.884|0.800|1.000|0.932|
> > > |Gemma2-9B|0.941|0.938|0.901|0.911|0.847|0.941|1.000|

---

> > > ### Author Response · Authors · 2025-07-07
> > > **Response to follow up comments (part 2)**
> > >
> > > > Other baselines
> > >
> > > We appreciate the suggestion. As suggersted, we conduct additional comparison and report results of Majority and Random baselines below, as well as of non-uncertainty based approach. Since SelfCheckGPT is a sentence-level approach (sampling responses and check consistency in sentence level) that requires a careful redesign to apply to our entity-level data, we do not report its result. Alternatively, for the non-uncertainty based approach, we prompt GPT-4o with the biography and the target entity, as ask if the entity is hallucination or not. The prompt is listed below:
> > >
> > > > You are a helpful and precise assistant for detect hallucinated entities. We would like to request your help on detecting hallucinations in a generated biography.
> > > >
> > > > We will give you a passage and a target entity in the passage. Please output False if the entity is a hallucination and otherwise True. You do not need to output anything else nor justify your answer.
> > >
> > > |Baseline|AUROC|AUPRC|Precision|Recall|F1|
> > > |---|---|---|---|---|---|
> > > |Majority|0.500|0.575|0|0|0|
> > > |Random|0.503|0.219|0.156|0.155|0.155|
> > > |GPT-4o|0.435|0.234|0.104|0.251|0.147|
> > >
> > > The result shows that all the uncertainty-based approaches outperform these baselines.
> > >
> > >
> > >
> > > > OOD data
> > >
> > > We have given serious consideration to the reviewer’s insightful question, and we fully agree that understanding how uncertainty-based detectors perform under distribution shift is an important direction. However, we found it very difficult to operationalize this idea in our current setting. The core challenge is that it is unclear how to define or construct OOD data in a principled way for our task. Since the pre-training data for both the target and proxy models are not fully disclosed or standardized, we cannot accurately quantify or control their distributional differences. Without a clear understanding of how their training distributions diverge, it is difficult to design meaningful OOD conditions or interpret any observed performance degradation.
> > >
> > > Moreover, unlike classic OOD settings where the input domain shift is explicit (e.g., digit recognition across MNIST and SVHN), uncertainty-based hallucination detection relies on language models' internal representations of semantics, reasoning, and world knowledge. This makes it difficult to define what kind of shift would count as "OOD" in a meaningful way. Potential shifts--such as changes in question format, topic domain, or factual grounding--might affect the proxy model's familiarity with the task, but there is no principled framework to measure or categorize these shifts as "OOD." Defining this precisely is non-trivial and merit dedicated investigation on its own.
> > >
> > > We appreciate the reviewer for raising this point and will incorporate this more explicitly in our discussion/limitation section.

---

### Review · Reviewer_XtDP · 2025-05-28

**Summary Of Contributions:**

This work introduces a novel dataset for entity-level hallucination detection. Leveraging this dataset, the authors evaluate 17 different large language models and explore 5 uncertainty scoring methods to assess their effectiveness in identifying hallucinated entities. The paper presents both quantitative and qualitative analyses of the experimental results, offering insights into the relationship between hallucination tendencies and various linguistic properties.

**Audience:**

Yes

**Claims And Evidence:**

Yes

**Requested Changes:**

1. **Clarify the entity-level labeling pipeline**:
I am unclear about the process of generating labels for entities in Section 3.1. My understanding is that the authors use FActScore to extract facts from the original text, assign each fact a ‘True’ or ‘False’ label, and then propagate those labels to entities within the facts. I have several questions about this process:
- In section 3.1, why is the fact “He was born on March 9, 1941” labeled as True while “He was born in Ramos Mejía” is labeled as False? Both seem supported by the original text. If the latter reflects hallucinated content, perhaps highlighting it in Figure 1 would clarify matters for readers. As presented, both facts appear correctly generated and factually plausible at first glance. Please also explain how fact labels are determined in the main text.
- The authors state that “*neutral words like ‘he’ and ‘on’ are considered non-hallucinated*,” but this may have limitations. For example, pronouns such as ‘he’ could themselves be hallucinated in gender-sensitive contexts (e.g., referring to a female subject as ‘he’ or ‘Mr.’). It would be helpful to discuss this potential limitation.

2. In second paragraph of Section 3.1, "*we leverage biographies generated by ChatGPT (OpenAI, 2023).*" For reproducibility, please specify which GPT model was used (e.g., GPT-4, GPT-3.5).
3. At the end of Section 3.1, the authors mention that you “*manually verify and refine to ensure correctness.*” **Could you elaborate on this verification process**? For instance:
- Who conducted the manual verification (e.g., recruited annotators or expert reviewers)?
- What guidelines or criteria were applied?
- Were any quantitative metrics recorded (e.g., correctness rate or inter-annotator agreement)?
4. I feel the dataset’s quality depends heavily on FActScore’s fact-checking and on Wikipedia’s coverage. Is FActScore’s knowledge updated to reflect current information? Additionally, since Wikipedia may not cover every relevant fact, how do you address potential false negatives? I would appreciate the authors’ perspective on these points.

**Strengths And Weaknesses:**

**Strength**: This work conducts extensive experiments across 17 large language models and five uncertainty-based hallucination detection methods. The authors provide both quantitative and qualitative analyses, offering valuable insights into model behavior and hallucination patterns.

**Weaknesses**: The dataset construction process is a little bit unclear, particularly how the labels for entities are obtained. The dataset's quality is not empirically validated, as the labeling is solely performed by GPT-4o. While the authors mention manual verification, the results of this process (e.g., user testing) are not presented in the paper. Finally, the study focuses exclusively on uncertainty-based hallucination detection methods, which limits the scope of the investigation.

---

> ### Author Response · Authors · 2025-07-02
>
> We sincerely appreciate your feedback and insightful comments! We address the questions and comments below in detail.
>
> > Why is the fact “He was born on March 9, 1941” labeled as True while “He was born in Ramos Mejía” is labeled as False
>
> Sorry for the confusion that we put different examples in Figure 1 and in the paragraph. We've changed the example in the paragraph to make it align with Figure 1.
>
> > Pronouns such as ‘he’ could themselves be hallucinated in gender-sensitive contexts
>
> In biography generation, pronouns usually refer to the person specified in the prompt. Therefore, we assume the pronouns themselves are always correct and consider other non-supported information related to the pronouns as a hallucination. Additionally, in our data verification process, we observed that GPT3.5 usually generates consistent pronouns for each biography, suggesting that it is less likely to hallucinate gender issues while generating a biography.
>
> > ChatGPT version for biography generation
>
> The responses are generated by GPT3.5.
>
> > Verification process
>
> The verification was conducted by authors of the paper, who are well familiar with hallucination detection and LLM.
>
> To verify the GPT-4o annotation, we provided the generated biography and original FActScore annotation as references. The verification includes two parts: 1) whether a sentence is segmented into reasonable entities. For example, "December 18, 1949" should be one entity instead of three. 2) whether the label of an entity aligns with the original FActScore annotation. For example, given an original annotation, "Lanny Flaherty was born on December 18, 1949 (non-support)," the entity, "December 18, 1949," should be labeled as False.
>
> The authors verified the whole dataset independently. When the discrepancy between the verification results exists, the annotators would discuss it to reach the final consensus.
>
> The Jaccard similarity between the GPT-4o annotation and the annotation after verification is 0.945, and the Cohen's Kappa inner-annotator agreement on entities' label (True/False) where the entities are the same before/after verification is 0.940. This result suggests that the GPT-4o annotation is reliable.
>
> > FActScore knowledge and quality
>
> FActScore reported a 90% inner-annotator agreement rate on labeling GPT3.5 generated biographies (Appendix A in [1]), suggesting high-quality labels. However, we agree that there are errors in the dataset as it verified claims based on Wikipedia. Since Wikipedia does not contain all the information to verify a biography, some factual claims may be missclassified as "non-support". In fact, in Appendix A of their paper, they reported that 10% of claims about people whose categories are either 'rare' or 'very rare' were misclassified. During our GPT-4o annotation process, we observed cases where the FActScore label was "non-support" while GPT-4o annotated the corresponding entity as "True." For these conflict cases (<5%), we manually verified them through Google search and updated the labels if needed.
>
> [1]: Min, Sewon et al., FACTSCORE: Fine-grained Atomic Evaluation of Factual Precision in Long Form Text Generation. 2023

---

> > ### Comment · Reviewer_XtDP · 2025-07-07
> > **Thank the authors for their responses**
> >
> > I have read their response and my concerns are addressed. The revised manuscript should reflects all the changes.

---

> > > ### Author Response · Authors · 2025-07-08
> > >
> > > We are glad that our response and revised manuscript have addressed all of your concerns. Thanks again for your time and effort!

---

### Review · Reviewer_cHGr · 2025-06-20

**Summary Of Contributions:**

The paper introduces HalluEntity, a novel dataset with 18,785 entity-level annotations based on biographies generated by ChatGPT, enriched with labels derived from atomic facts. It focuses on the challenge of fine-grained hallucination detection in large language models (LLMs) outputs. They evaluate five uncertainty-based methods—Likelihood, Entropy, CCP, SAR, and Focus—across 17 modern LLMs. The findings reveal that token-level uncertainty measures tend to over-predict hallucinations, while context-aware approaches (SAR, Focus) perform better but still struggle to maintain balanced precision and recall, particularly in low-hallucination scenarios.

**Audience:**

Yes

**Claims And Evidence:**

No

**Requested Changes:**

1. Please ensure the annotation process has multiple runs from the same or multiple annotators (LLMs).
2. Please list inter-annotator agreements among multiple annotators (LLMs).
3. Is there any relation between FPR/FNR and Entropy?
4. The study notes that some scores assign high uncertainty to informative content like named entities, leading to false positives. What makes them "uncertain" to the model, even when correct? Is it low frequency in the training data, or something more complex?

**Strengths And Weaknesses:**

Strengths:
1. Introduces the first large-scale, entity-level hallucination detection dataset with real LLM generations.
2. Evaluates 5 distinct uncertainty-based methods over 17 LLMs, analyzing the effect of model family and size.
3. The paper includes detailed qualitative and quantitative analysis.

Weaknesses
1. The paper only focuses on the performance of biography.
2. While efficient, the annotation introduces the risk of propagation of GPT-4o’s own hallucination tendencies or missegmentations. There are no multi-agreements among multiple annotators (LLMs) or multiple runs of the same annotator (LLM).
3. Also, the entity labeling process is derived from the atomic fact labels in the FACTSCORE dataset. Any biases or errors in the original FACTSCORE annotations would propagate into HALLUENTITY.

---

> ### Author Response · Authors · 2025-07-02
>
> We sincerely appreciate your feedback and insightful comments! We address the questions and comments below in detail.
>
> > Focus on biography only
>
> We chose the biography domain intentionally as a starting point for fine-grained hallucination detection because it is rich in named entities and has well-documented ground truth via Wikipedia, which enables scalable and verifiable annotation at the entity level. This setting is especially relevant since many real-world LLM applications (e.g., digital assistants, educational content) involve biographical or factual content where hallucinated entities can be particularly problematic. That said, we acknowledge that the dataset will be more impactful if it covers a wide range of tasks. We view this work as establishing a foundation for broader investigations, and future extensions to news domains are possible with appropriate reference corpora. We have mentioned this issue in the limitation section.
>
> > Multiple-run annotation, inter-annotator agreement
>
> We performed three run annotations for a random subset of samples. We then compute the Jaccard similarity across the three entity sets. The similarity score is 0.813, suggesting a reasonably high similarity. We further compute the Fleiss's Kappa inner-annotator agreement on entities' label (True/False) where the entities are the same across three runs. The Fleiss's Kappa across the three runs is 0.984, suggesting a near-perfect agreement.
>
> > Biases/errors in FActScore dataset
>
> FActScore reported a 90% inner-annotator agreement rate on labeling GPT3.5 generated biographies (Appendix A in [1]), suggesting high-quality labels. However, we agree that there are errors in the dataset as it verified claims based on Wikipedia. Since Wikipedia does not contain all the information to verify a biography, some factual claims may be missclassified as "non-support." In fact, in Appendix A of their paper, they reported that 10% of claims about people whose categories are either 'rare' or 'very rare' were misclassified. During our GPT-4o annotation process, we observed cases where the FActScore label was "non-support" while GPT-4o annotated the corresponding entity as "True." For these conflict cases (<5%), we manually verified them through Google search and updated the labels if needed.
>
> > Relation between FPR/FNR and Entropy
>
> Entropy has a similar trend of FPR/FNR across POS and NER tags compared to Likelihood, with a higher FPR but lower FNR. This suggests that similar to Likelihood, Entropy tends to over-predict hallucination and can not precisely locate hallucinated content.
>
> > Uncertain but correct generation
>
> When generating biographies, correct information sometimes can be generated in different order. For example, in the generation, "*She later transitioned to acting, appearing in films such as "A Bronx Tale" (1993), "Just Cause" (1995), and "Belly" (1998),*" the list of films can be generated in different orders while preserving the factuality. In this case, there could be multiple correct next tokens of "appearing in films such as" and thus each of them will get a small probability.
>
> [1]: Min, Sewon et al., FACTSCORE: Fine-grained Atomic Evaluation of Factual Precision in Long Form Text Generation. 2023

---

### Author Response · Authors · 2025-07-02
**Changes Since Last Submission**

We thank all reviewers for their insightful feedback. We have revised the paper accordingly, and the major changes are summarized below:

- **[R1]** Add limitation about biography only
- **[R1]** Explain the scenario of uncertain but correct generation in Sec. 5.3
- **[R1]** Add a description of the relation between FPR/FNR and Entropy to Sec. 5.3
- **[R1, R2]** Add limitation about errors/biases inhered from FActScore
- **[R1, R2]** Add details of data quality assessment and report the result of multiple-run annotation and inner-annotator agreement to Appendix A
- **[R2]** Modify the example in 3.1 to align with Figure 1
- **[R3]** Change the notations of entities, token-level labels, and entity-level labels
- **[R3]** Remove the 2nd y-axis in Figure 2 and put percentage value inside the bars
- **[R3]** Add discussion about proxy LLM

All the added content has been marked as red.
(* We refer to Reviewer cHGr as R1, Reviewer XtDP as R2, and Reviewer 3S6k as R3.)

---

### Decision · Action_Editor_agZw · 2025-09-03

**Recommendation:** Accept as is

**Audience:**

Yes

**Audience Explanation:**

The paper is highly topical in its large-scale consideration of entity hallucination in LLMs. This, together with the new dataset/benchmark and the thoroughness of the evaluation make it a valuable contribution for the community.

**Claims And Evidence:**

Yes

**Claims Explanation:**

The paper investigates entity-level hallucination tendencies of large language models (LLMs). To that end, it introduces HalluEntity, a biography dataset with entity-level annotations that is intended to support fine-grained hallucination detection in LLM outputs. Using HalluEntity, the paper evaluates five uncertainty-based methods---likelihood, entropy, claim-conditioned probability (CCP), shifting attention to relevance, and focus---across 17 recent LLMs to assess their effectiveness in identifying hallucinated entities. The study finds that token-level uncertainty scores tend to over-predict hallucinations, while context-aware methods perform better but still face challenges, especially in low-hallucination settings. Detailed linguistic analyses further reveal how hallucination patterns vary with part-of-speech, named entity type, and sentence position.

The primary claims of the paper involve the relationship between token-level uncertainty and LLM hallucinations. All three reviewers agree that the paper is thorough in this regard and they emphasize their appreciation for the detailed quantitative and qualitative analysis. The reviewers initially raised concerns regarding the paper's focus on the uncertainty of external models rather than the token-level uncertainty of the target model; the risk that the hallucination tendencies of the underlying LLM might be propagated; and questions of clarity and quality of the proposed dataset. As Reviewer 3S6k comments, the responses that the authors provided, together with their revisions to the paper, helped to resolve these concerns.